# Antifibrotic Effects of an α7 Nicotinic Acetylcholine Receptor Agonist in Keloid Fibroblasts and a Rat Scar Model

**DOI:** 10.3390/ijms26188868

**Published:** 2025-09-11

**Authors:** Hyun Roh, Yo Han Kim, Kyung Jun Heo, Jong Won Hong, Won Jai Lee

**Affiliations:** Department of Plastic and Reconstructive Surgery, Yonsei University, 50-1 Yonsei-Ro, Seodaemun-Gu, Seoul 03722, Republic of Korea; nh1005@yuhs.ac (H.R.); yhyh136@yuhs.ac (Y.H.K.); hkj9061@yuhs.ac (K.J.H.); saturn@yuhs.ac (J.W.H.)

**Keywords:** α7 nicotinic acetylcholine receptor, keloid fibroblasts, tropisetron, fibrosis, inflammation, TGF-β/Smad signaling, NF-κB, extracellular matrix

## Abstract

Keloids are characterized by excessive extracellular matrix (ECM) accumulation and persistent inflammation, leading to disfiguring scars and poor therapeutic outcomes. The α7 nicotinic acetylcholine receptor (α7nAChR) has emerged as a key modulator of inflammatory and fibrotic signaling. This study evaluated the antifibrotic effects of tropisetron, a clinically available α7nAChR agonist, in keloid fibroblasts (KFs) and a rat incisional scar model. In vitro, KFs exhibited reduced α7nAChR expression, which was restored by tropisetron in a dose-dependent manner. Tropisetron treatment significantly decreased KF viability, downregulated pro-fibrotic genes (COL1A1, COL3A1, α-SMA), and upregulated matrix metalloproteinases (MMP1 and MMP3). Additionally, it suppressed phosphorylation of Smad2/3 and reduced expression of NF-κB and TNF-α, indicating inhibition of both TGF-β and inflammatory pathways. In vivo, tropisetron-treated rats showed a ~40% reduction in scar area, improved collagen organization, and increased α7nAChR expression in scar tissue. Western blot analysis confirmed decreased levels of collagen I, p-Smad2/3, α-SMA, NF-κB, and TNF-α. These results indicate that tropisetron exerts dual antifibrotic and anti-inflammatory effects through α7nAChR-mediated signaling and enhanced ECM remodeling. This study provides the first evidence supporting α7nAChR activation as a promising therapeutic strategy for managing keloids and other fibrotic skin disorders.

## 1. Introduction

Fibrosis is a fundamental biological process involved in tissue repair, primarily driven by fibroblast activation and excessive deposition of extracellular matrix (ECM), especially collagen. While this process is essential for wound healing, dysregulated fibrosis can lead to serious diseases such as cirrhosis, pulmonary fibrosis, and glomerulonephritis, where abnormal accumulation of fibrous tissue impairs organ function [1,2]. In the skin, fibrosis typically presents as scarring, including hypertrophic scars and keloids—the latter being particularly resistant to conventional treatments [3]. Although keloids have been extensively studied, their pathogenesis remains incompletely understood and is believed to involve chronic inflammation that sustains fibroblast activation and ECM overproduction [4,5]. Keloids are characterized by rapid proliferation of fibroblasts and excessive ECM deposition, exhibiting tumor-like behavior that disrupts normal tissue architecture and function [6]. Understanding the mechanisms underlying keloid formation is critical for identifying novel therapeutic targets. Among the factors implicated in fibrotic pathogenesis, the nicotinic acetylcholine receptor (nAChR), particularly the α7 subtype (α7nAChR), has emerged as a potential regulator of inflammation and fibrosis [7,8].

The α7nAChR is a ligand-gated ion channel expressed in both neuronal and non-neuronal tissues. It exerts anti-inflammatory effects via the cholinergic anti-inflammatory pathway, primarily by inhibiting the release of proinflammatory cytokines such as TNF-α, IL-6, and IL-1β [9,10,11]. In addition to its immunomodulatory role, α7nAChR activation influences cell proliferation, apoptosis, and differentiation, and has demonstrated therapeutic potential in various inflammatory and fibrotic conditions [12,13,14]. Notably, α7nAChR signaling has been linked to suppression of key fibrotic pathways, including the TGF-β/Smad and NF-κB signaling cascades [15,16,17].

Recent studies have further highlighted the antifibrotic potential of α7nAChR. Selective agonists such as GTS-21 and PHA-543613 have been shown to inhibit ECM production and fibrosis in experimental models of pulmonary and dermal fibrosis [18,19]. Tropisetron, a clinically approved 5-HT3 receptor antagonist commonly used as an antiemetic to treat chemotherapy-induced nausea and vomiting, also acts as an α7nAChR agonist. It has demonstrated anti-inflammatory and antifibrotic effects in multiple tissues, including in models of skin and lung fibrosis [20,21,22]. In models of skin fibrosis, tropisetron has been shown to reduce collagen deposition and restore normal tissue architecture, potentially through modulation of oxidative stress and fibrogenic signaling pathways [23,24]. Moreover, α7nAChR signaling has been reported to attenuate fibroblast-to-myofibroblast differentiation—a hallmark of fibrotic progression in both keloids and hypertrophic scars [25,26]. Its expression also appears to be dynamically regulated during wound healing, further supporting its possible role in scar modulation [27,28].

Collectively, these findings suggest that α7nAChR activation may suppress fibrotic responses by modulating inflammatory and profibrotic pathways. However, its specific role in keloid pathogenesis remains poorly understood. To date, no transcriptomic or proteomic datasets have reported α7nAChR (CHRNA7) expression levels in keloid fibroblasts. In this study, we provide the first experimental evidence of α7nAChR downregulation in KFs using immunocytochemistry, qRT-PCR, and Western blotting. This novel finding has also been filed as a national patent, supporting its originality and potential therapeutic relevance. Therefore, the present study aimed to investigate the expression and function of α7nAChR in human keloid fibroblasts (KFs) and keloid tissues and to evaluate the antifibrotic effects of tropisetron in both in vitro and in vivo models. These findings may provide new insights into α7nAChR as a potential therapeutic target for keloid treatment.

## 2. Results

### 2.1. In Vitro Study

#### 2.1.1. Expression of α7nAChR in HDFs and KFs

To evaluate the basal expression of α7nAChR in normal and pathological conditions, immunocytochemistry, qRT-PCR, and Western blot analyses were performed using HDFs, LPS-treated HDFs, and KFs. The expression of α7nAChR was compared among normal human dermal fibroblasts (HDFs), HDFs treated with lipopolysaccharide (LPS), and KFs using immunocytochemistry, quantitative reverse transcription polymerase chain reaction (qRT-PCR), and Western blot analysis (Figure 1). Immunofluorescence staining revealed reduced cytoplasmic red fluorescence intensity in both LPS-treated HDFs and KFs compared to untreated HDFs, indicating a decrease in α7nAChR expression. (Figure 1a). qRT-PCR analysis demonstrated a significant decrease in α7nAChR mRNA expression in the HDF + LPS and KF groups compared to the HDF group (* *p* < 0.05), with no statistically significant difference between HDF + LPS and KF groups (Figure 1b). Western blot analysis further confirmed a marked reduction in α7nAChR protein levels in both HDF + LPS and KF groups relative to HDFs (** *p* < 0.01), again showing no significant difference between the two (Figure 1c). These findings suggest that α7nAChR mRNA and protein expression are significantly suppressed under inflammatory conditions and in KFs.

#### 2.1.2. Tropisetron Enhances α7nAChR Expression in KFs

To assess whether tropisetron modulates α7nAChR expression in keloid fibroblasts, KFs were treated with various concentrations of tropisetron followed by immunocytochemistry and qRT-PCR analysis. KFs were treated with tropisetron, a selective α7nAChR agonist, at concentrations of 10 and 20 μg/mL, to examine changes in α7nAChR expression and function. Immunocytochemical staining confirmed enhanced α7nAChR expression in tropisetron-treated KFs, as indicated by increased red fluorescence intensity compared to untreated controls (Figure 2a). qRT-PCR analysis further demonstrated that tropisetron increased α7nAChR mRNA expression in a dose-dependent manner (Figure 2b). These findings suggest that tropisetron upregulates α7nAChR expression in KFs, potentially enhancing its functional activity. Although α7nAChR is typically a membrane-bound receptor, some intracellular staining was observed, likely due to permeabilization during immunocytochemistry, which allows antibody access to both surface and intracellular compartments.

#### 2.1.3. Effects of Tropisetron on KF Viability and Fibrotic Gene/Protein Expression

To determine the antifibrotic potential of tropisetron in keloid fibroblasts, we conducted MTT assays, qRT-PCR, and Western blot analyses following treatment with increasing concentrations of tropisetron. Specifically, cell viability, fibrotic gene expression, and protein levels were evaluated following treatment with 1, 10, and 20 µg/mL of tropisetron. MTT assays showed a significant, dose-dependent reduction in KF viability following tropisetron treatment compared to untreated controls (Figure 3a). Viability decreased significantly at 10 μg/mL (* *p* < 0.05) and 20 μg/mL (** *p* < 0.01). qRT-PCR analysis revealed that fibrotic markers, including collagen I, collagen III, and α-smooth muscle actin (α-SMA), were markedly upregulated in KFs compared to HDFs, and were dose-dependently downregulated following tropisetron treatment. In contrast, expression of matrix metalloproteinases (MMP1) and MMP3, which were reduced in KFs, were increased after tropisetron treatment (Figure 3b). Western blot analysis confirmed these findings at the protein level. Elevated collagen I expression in KFs was significantly reduced at both 10 and 20 μg/mL (** *p* < 0.01), with no significant difference between the two doses. α-SMA expression, which was significantly elevated in KFs compared to HDFs (** *p* < 0.01), was not significantly reduced by 10 μg/mL tropisetron but was significantly reduced at 20 μg/mL (* *p* < 0.05 vs. KF). Phosphorylated Smad2/3 (p-Smad 2/3) levels were significantly reduced in both treatment groups compared to untreated KFs (** *p* < 0.01), with no additional reduction at 20 μg/mL (Figure 3c,d). To further investigate the underlying signaling mechanisms, Western blot analysis was performed on KFs treated with 10 and 20 μg/mL tropisetron. The results revealed that tropisetron suppressed NF-κB expression, which was associated with a concomitant decrease in TNF-α expression (Figure 3e). Quantitative densitometric analysis confirmed a dose-dependent reduction in both NF-κB and TNF-α expression (Figure 3f). These results suggest that tropisetron exerts antifibrotic effects by downregulating NF-κB-mediated cytokine signaling, thereby reducing TNF-α expression.

### 2.2. In Vivo Rat Incisional Scar Model

#### 2.2.1. Evaluation of the Antifibrotic Effect of Tropisetron

To explore the in vivo antifibrotic effect of tropisetron, we utilized a rat incisional wound model. Scar tissue was examined using immunohistochemistry and histological staining techniques. To evaluate the drug’s efficacy in reducing scar formation, wound tissues were collected and subjected to histological analysis 14 days after surgery. The experimental groups included phosphate-buffered saline (PBS)-treated controls and tropisetron-treated rats (20 μg; *n* = 5 per group). As shown in Figure 4a, α7nAChR expression was markedly increased in the dermal layer of the tropisetron-treated group compared to controls. Positive staining was more extensive and intense, indicating upregulation of α7nAChR in the scar tissue. Quantitative image analysis using MetaMorph® image analysis software (Molecular Devices, San Jose, CA, USA) (Figure 4b) demonstrated a statistically significant increase in the α7nAChR-positive area in the tropisetron-treated group (* *p* < 0.05).

Hematoxylin and eosin (H&E) staining (Figure 4c) revealed that the control group exhibited extensive scar areas, dense inflammatory cell infiltration, and disorganized dermal structure. In contrast, the tropisetron-treated group showed markedly reduced inflammatory infiltration and improved dermal organization. Masson’s trichrome (MT) staining (Figure 4d) confirmed these findings, demonstrating densely packed collagen bundles in the control group, whereas tropisetron-treated tissues exhibited reduced collagen deposition and thinner, more loosely arranged collagen fibers. Quantitative analysis of scar area (Figure 4e) revealed a significant reduction in the tropisetron-treated group compared to PBS-treated controls (** *p* < 0.01), suggesting that tropisetron effectively promotes scar remodeling and reduces fibrosis in vivo. These findings indicate that tropisetron upregulates α7nAChR expression in vivo and supports the antifibrotic and anti-inflammatory effects observed in vitro.

#### 2.2.2. Tropisetron Reduces Fibrotic and Inflammatory Signaling in Incisional Wound Tissue

To elucidate the mechanisms behind tropisetron’s in vivo effects, we performed Western blot analysis to examine the expression of fibrosis- and inflammation-related proteins in scar tissues. Specifically, scar tissues were collected from PBS-treated (control) and tropisetron-treated rats (20 μg) at 14 days post-surgery (*n* = 5 per group), and Western blotting was conducted to analyze fibrotic and inflammatory marker expression levels. As shown in Figure 5a, the expression of collagen I, α-SMA, and p-Smad 2/3 were markedly reduced in the tropisetron-treated group compared with the controls. Densitometric analysis (Figure 5c) confirmed that collagen I (*p* < 0.001), α-SMA (*p* < 0.05), and p-Smad2/3 (*p* < 0.05) were significantly downregulated. These results suggest that tropisetron suppresses key fibrotic signaling pathways, particularly TGF-β/Smad signaling, within the scar microenvironment. Western blot analysis of inflammatory mediators (Figure 5b) revealed that NF-κB and TNF-α expression levels were also significantly reduced in tropisetron-treated scars compared to controls. Quantification (Figure 5d) showed significant reductions in NF-κB (*p* < 0.01) and TNF-α (*p* < 0.05) expression. These findings indicate that tropisetron exerts potent anti-inflammatory effects in vivo, which may contribute to improved wound remodeling and reduced scar formation. Together, these data demonstrate that tropisetron suppresses fibrosis and inflammation in vivo by downregulating ECM deposition and proinflammatory signaling pathways in keloid-like scar tissue.

## 3. Discussion

Keloids are pathological scars that result from aberrant wound healing characterized by excessive fibroblast proliferation and ECM accumulation. A central contributor to this fibrotic process is the prolonged activation of the TGF-β/Smad signaling pathway, along with persistent inflammatory responses mediated by cytokines and immune cell interactions [1,2,3,4]. Recent studies have highlighted the regulatory role of α7nAChR in modulating inflammation and fibrosis across various tissues, including dermal repair [7,8,9]. Activation of α7nAChR has been shown to inhibit proinflammatory cytokine release and suppress fibroblast-driven ECM deposition, suggesting its therapeutic potential as an antifibrotic modulator [10,11,12]. However, the mechanistic role of α7nAChR in keloid pathogenesis remains largely unexplored.

Previous studies have demonstrated that activation of α7nAChR can attenuate fibrotic responses in various organs, including the lungs, liver, and heart, by suppressing key pro-fibrotic signaling pathways [10,11,13,18]. In this context, our study provides the first experimental evidence that tropisetron, a clinically approved α7nAChR agonist, effectively restores α7nAChR expression in KFs and significantly downregulates major fibrotic markers—including collagen I, collagen III, and α-SMA—in a pathologic dermal environment. These findings suggest that α7nAChR signaling plays a crucial antifibrotic role in keloid formation and highlight tropisetron as a potential therapeutic agent for managing cutaneous fibrosis.

It has been well established that suppressing fibroblast proliferation and downregulating collagen synthesis are essential to preventing excessive scar formation [19,21]. Consistent with this, prior studies have shown that tropisetron suppresses collagen synthesis in dermal fibroblasts via α7nAChR activation in models of scleroderma [18]. Similarly, our data demonstrate that tropisetron significantly reduced KF viability in a dose-dependent manner and suppressed fibrotic gene and protein expression, including collagen I, collagen III, and α-SMA. These results suggest that tropisetron may restore homeostatic regulation of ECM production and inhibit fibrotic activation in KFs.

MMPs, particularly MMP1 and MMP3, play a central role in ECM degradation and tissue remodeling. Their reduced expression is a hallmark of keloid pathology [6,20,29,30]. Notably, α7nAChR activation has been shown to enhance MMP expression in fibrotic models, such as lung fibrosis, promoting ECM turnover [31,32]. In our study, tropisetron treatment significantly increased the expression of MMP1 and MMP3 in KFs, indicating enhanced ECM degradation. These findings suggest that tropisetron may contribute to scar remodeling and mitigate fibrotic burden by restoring MMP activity in keloid tissues.

Previous studies have shown that cholinergic signaling—particularly via α7nAChR activation—facilitates the resolution of inflammation and promotes regenerative healing in fibrotic skin disorders [8,20,25]. Consistent with these findings, histological evaluation of scar tissues in our rat model revealed that tropisetron-treated animals exhibited reduced inflammatory cell infiltration and more organized collagen architecture compared to controls, indicating enhanced tissue remodeling and scar maturation. In addition to α7nAChR, other endogenous cholinergic regulators such as SLURP-2 have been implicated in cutaneous wound healing. Recent studies reported that SLURP-2 mimetic peptides enhance keratinocyte viability and migration, contributing to re-epithelialization and tissue regeneration through nicotinic receptor-mediated signaling [33,34]. Although the cellular targets of SLURP-2 may differ from tropisetron, both agents appear to share convergent pathways involving nicotinic receptor activation and wound repair facilitation. Further comparative studies are warranted to delineate distinct versus overlapping roles of SLURP-2 and α7nAChR-targeting agents in dermal remodeling.

The observed upregulation of α7nAChR expression in scar tissues following tropisetron treatment suggests that this receptor may play a regulatory role in fibrosis resolution. Prior studies have demonstrated that α7nAChR activation exerts anti-inflammatory and antifibrotic effects by modulating immune cell–fibroblast interactions and suppressing pro-fibrotic signaling cascades, including TGF-β/Smad pathway [9,14,19]. Our findings support this concept and highlight α7nAChR as a potential therapeutic target in pathological scarring. However, this study primarily assessed α7nAChR expression through immunocytochemistry and gene expression analyses. Although these methods demonstrated increased α7nAChR levels following tropisetron treatment, we did not directly examine membrane-specific localization. Given the functional importance of membrane-bound α7nAChR in mediating ligand interactions, future studies including membrane fraction analysis or surface biotinylation assays are needed to validate the receptor’s cellular distribution.

Persistent activation of TGF-β/Smad and NF-κB signaling pathways has been widely implicated in keloid pathogenesis, where it promotes fibroblast activation and proinflammatory cytokine production [15,16,17]. In this study, tropisetron treatment significantly suppressed p-Smad2/3, NF-κB, and TNF-α expression in both KFs and scar tissue, supporting the hypothesis that inhibition of these pathways attenuates fibrotic progression. These findings suggest that tropisetron exerts dual antifibrotic and anti-inflammatory effects by downregulating key pro-fibrotic and proinflammatory signaling axes. While tropisetron is a known partial agonist of α7nAChR, it can also interact with other receptors such as 5-HT_3_. Therefore, the possibility of off-target effects cannot be completely ruled out. However, the observed antifibrotic and anti-inflammatory effects in this study closely align with known α7nAChR-mediated mechanisms reported in previous studies [15,32]. Future studies incorporating α7nAChR-specific inhibitors such as α-bungarotoxin or gene silencing approaches will be necessary to confirm the receptor-specific actions of tropisetron.

The anti-inflammatory actions of α7nAChR agonists are well-documented, particularly their ability to inhibit NF-κB nuclear translocation and reduce proinflammatory cytokine expression including TNF-α [7,20,22]. Consistent with these mechanisms, our study demonstrated that tropisetron reduced NF-κB and TNF-α expression in both KFs and scar tissues, suggesting that inhibition of this pathway contributes to the observed anti-inflammatory and antifibrotic effects.

This study aimed to elucidate the role of α7nAChR in keloid pathogenesis and to evaluate the antifibrotic efficacy of tropisetron, an α7nAChR agonist. In vitro analyses demonstrated that tropisetron restored α7nAChR expression in KFs, suppressed fibrotic markers—including collagen I, collagen III, and α-SMA, and reduced inflammatory cytokine expression. Tropisetron also enhanced ECM degradation by upregulating MMP1 and MMP3, and inhibited activation of TGF-β/Smad and NF-κB signaling pathways. In vivo, tropisetron treatment increased α7nAChR expression in scar tissues and improved histological features, including reduced scar size, reduced inflammatory cell infiltration, and more organized collagen architecture. Together, these findings suggest that modulation of α7nAChR may attenuate fibrotic progression by simultaneously regulating inflammation and ECM remodeling. Tropisetron, through its ability to modulate this pathway, emerges as a promising candidate for treatment of cutaneous fibrosis such as keloids.

Given its clinical use as an antiemetic, tropisetron holds potential for rapid translation into keloid therapy with minimized development costs.

Further studies using human tissue models and clinical validation are necessary to confirm the therapeutic applicability of α7nAChR-targeted strategies in pathological scarring.

## 4. Materials and Methods

### 4.1. In Vitro Studies

#### 4.1.1. Cell Culture

Normal HDFs and KFs were obtained from the American Type Culture Collection (ATCC, Manassas, VA, USA). Cells were cultured in Dulbecco’s Modified Eagle Medium (DMEM; ATCC) supplemented with 10% fetal bovine serum (FBS), 100 U/mL penicillin, and 100 μg/mL streptomycin, and maintained at 37 °C in a humidified incubator containing 5% CO_2_.

#### 4.1.2. α7nAChR Activation Assays

To evaluate the effects of α7nAChR activation, HDFs and KFs were treated with LPS and various concentrations of tropisetron. Tropisetron (C_18_H_21_N_3_O_2_) is a water-soluble small molecule known as a selective partial agonist of α7 nicotinic acetylcholine receptor (α7nAChR), originally developed as a 5-HT_3_ receptor antagonist. Cell viability was assessed using the MTT assay. Gene and protein expression levels were analyzed by qRT-PCR and Western blotting, respectively. Immunofluorescence staining and immunoprecipitation were performed to assess protein localization and interaction.

#### 4.1.3. Immunofluorescence Assay

Cells were washed twice with PBS, fixed in 4% paraformaldehyde for 20 min at room temperature, and washed three times with PBST (PBS containing 0.01% Tween 20). After blocking with 5% bovine serum albumin (BSA) for 1 h, cells were incubated overnight at 4 °C with rabbit anti-α7nAChR primary antibody. The next day, cells were washed and incubated with Texas Red-conjugated bovine anti-mouse IgG secondary antibody (Santa Cruz Biotechnology, Dallas, TX, USA) for 2 h at room temperature. Nuclei were counterstained with 4′,6-Diamidino-2-phenylindole (DAPI) (Vector Laboratories, Burlingame, CA, USA), and samples were mounted for imaging using a confocal microscope (LSM700, Olympus Corp., Center Valley, PA, USA).

#### 4.1.4. Cell Viability Assay

Cell viability and metabolic activity were measured using the MTT assay. HDFs and KFs (5 × 10^4^ cells/cm^2^) were treated with 100 ng/mL LPS and tropisetron (1, 10, or 20 μg/mL) for 48 h. Cells were then incubated with 200 μL of 0.5 mg/mL MTT solution (Boehringer, Mannheim, Germany) for 3 h at 37 °C. After removing the MTT solution, 200 μL of dimethyl sulfoxide was added to dissolve formazan crystals. Absorbance was measured at 570 nm using an ELISA reader (Bio-Rad, Hercules, CA, USA).

#### 4.1.5. qRT-PCR

HDFs and KFs (2 × 10^5^ cells) were treated with LPS (100 ng/mL) and tropisetron (10 or 20 μg/mL) for 48 h. Total RNA was extracted using the RNeasy Mini Kit (Qiagen, Hilden, Germany), and cDNA was synthesized with the AccuPower RT PreMix First-Strand cDNA Synthesis Kit (Bioneer, Daejeon, Republic of Korea), according to the manufacturer’s instructions. Gene expression was analyzed using TaqMan primer/probe kits (Applied Biosystems) on an ABI Prism 7500 HT Sequence Detection System (Applied Biosystems, Foster City, CA, USA). The following primers were used: COL1A1 (Hs00164004_m1), COL3A1 (Hs00164103_m1), MMP1 (Hs00233958_m1), MMP3 (Hs00233962_m1), CHRNA7 (α7nAChR, Hs01010386_m1), ACTA2 (α-SMA, Hs00426835_g1), and GAPDH (Hs99999905_m1) as the internal control. For cDNA amplification, AmpliTaqGold DNA polymerase was activated by incubation at 95 °C for 10 min; followed by 40 cycles of denaturation at 95 °C for 15 s and annealing/extension at 60 °C for 1 min per cycle. The threshold cycle (Ct) was determined as the cycle at which fluorescence exceeded background levels. Relative mRNA expression was calculated using the 2^−ΔΔCt^ method, and values were normalized to GAPDH.

#### 4.1.6. Western Blot Analysis

Cells were lysed in RIPA buffer (Thermo Fisher Scientific, Waltham, MA, USA) supplemented with protease and phosphatase inhibitor cocktails (Sigma-Aldrich, St. Louis, MO, USA). Protein concentrations were determined using a BCA assay (Thermo Fisher Scientific), and 20 μg of total protein per sample was separated by 12% sodium dodecyl sulfate–polyacrylamide gel electrophoresis (SDS-PAGE) and transferred to polyvinylidene difluoride (PVDF) membranes (Millipore, Burlington, MA, USA). Membranes were blocked with 5% non-fat skim milk in TBST (Tris-buffered saline containing 0.1% Tween-20) for 2 h at room temperature, followed by overnight incubation at 4 °C with primary antibodies against α7nAChR (1:500, ab10096, Abcam, Cambridge, UK), collagen type I (1:1000, ab34710, Abcam), α-SMA (1:1000, ab7817, Abcam), p-Smad2/3 (Ser465/467, 1:1000, #8828, Cell Signaling Technology, Danvers, MA, USA), NF-κB p65 (1:1000, #6956, Cell Signaling Technology), TNF-α (1:1000, #3707, Cell Signaling Technology), and β-actin (1:5000, sc-47778, Santa Cruz Biotechnology, Dallas, TX, USA). After washing, membranes were incubated with horseradish peroxidase (HRP)-conjugated secondary antibodies (goat anti-mouse IgG, 1:2000, sc-2005, and goat anti-rabbit IgG, 1:2000, sc-2004; Santa Cruz Biotechnology) for 2 h at room temperature. Protein bands were visualized using enhanced chemiluminescence (ECL) system (Amersham, GE Healthcare, Chicago, IL, USA) and imaged with a ChemiDoc imaging system (Bio-Rad, Hercules, CA, USA). Band intensities were quantified using ImageJ software version 1.49 (National Institutes of Health, Bethesda, MD, USA), and protein expression levels were normalized to β-actin.

### 4.2. In Vivo Rat Incisional Scar Model

A rat incisional wound model was employed to evaluate the antifibrotic effects of tropisetron. Ten male Sprague–Dawley rats (8 weeks old, 250–300 g) were included in the study. All procedures were approved by the Institutional Animal Care and Use Committee (IACUC) of Yonsei University (IACUC No. 2021-0212). Anesthesia was induced via intraperitoneal injection of a combination of zolazepam-tiletamine (30 mg/kg, Zoletil^®^; Virbac, Carros, France) and xylazine (10 mg/kg, Rompun^®^; Bayer, Leverkusen, Germany). An 6 × 1 cm^2^ rectangle section of skin, subcutaneous fat, and muscle were excised with full thickness and only the skin layer was closed to maximize tension by leaving the muscle unsutured (Figure 6).

After surgery, animals were randomly assigned to two groups: the control group (Group C, *n* = 5), which received PBS (1 mL), and the tropisetron-treated group (Group T, *n* = 5), which received 20 μg of tropisetron diluted in 1 mL of PBS. Injections were administered intradermally into the wound margins using a 1 mL syringe fitted with a 27-gauge needle. Treatments were delivered on 0, 1, and 3 days after surgery. The injection sites were gently massaged to ensure even drug distribution.

#### 4.2.1. Histological Analysis

On postoperative day (POD) 14, all animals were euthanized, and wound tissues were harvested. Tissue specimens (5 mm thickness) were collected from the central region of each incisional scar, where skin tension was maximal. Samples were fixed in 4% paraformaldehyde, embedded in paraffin blocks, and sectioned at 5 μm thickness. Sections were stained with H&E and MT to evaluate general tissue morphology and collagen deposition, respectively. Scar area and granulation tissue were assessed under a light microscope at 40× magnification. To quantify scar formation, the boundary between the epidermis and the panniculus carnosus layer was measured. For each wound, two representative MT-stained sections were analyzed. Scar areas were calculated using ImageJ software (version 1.49; National Institutes of Health, Bethesda, MD, USA). Pixel-based measurements were converted to square micrometers using calibrated scale bars. Data are presented as mean ± SEM.

#### 4.2.2. Immunohistochemistry for α7nAChR

Paraffin-embedded sections (5 μm) were deparaffinized, rehydrated, and subjected to antigen retrieval in citrate buffer (pH 6.0) at 95 °C for 20 min. Endogenous peroxidase was blocked with 3% hydrogen peroxide, and nonspecific binding was blocked using 5% normal goat serum. Sections were incubated overnight at 4 °C with rabbit anti-α7nAChR primary antibody (ab10096, Abcam; 1:100). After washing, slides were incubated with biotin-conjugated goat anti-rabbit IgG (1:200, Vector Laboratories), followed by ABC reagent (Vector Laboratories). Diaminobenzidine (DAB) was used for color development, and slides were counterstained with hematoxylin. Images were captured using a bright-field microscope (Olympus BX51 or equivalent). α7nAChR-positive staining was quantified in five randomly selected high-power fields per section using MetaMorph^®^ software (Molecular Devices, San Jose, CA, USA). Data are expressed as the percentage of stained area relative to the total tissue area.

### 4.3. Statistical Analysis

Data were analyzed using one-way ANOVA for experiments involving more than two groups and a paired t-test for comparisons between two groups. Data are presented as means ± SEM, with *p* < 0.05 deemed statistically significant.

## 5. Conclusions

This study demonstrates that tropisetron, a selective α7nAChR agonist, effectively suppresses fibrosis and inflammation in KFs and a rat incisional scar model. Tropisetron exerted its effects by restoring α7nAChR expression, downregulating fibrotic markers, and modulating TGF-β/Smad and NF-κB signaling. These findings support the therapeutic potential of α7nAChR activation in modulating the fibrotic microenvironment and suggest tropisetron as a promising candidate for the treatment of keloids and other fibrotic skin disorders. Given its existing clinical approval as an antiemetic, repurposing tropisetron for cutaneous fibrosis could accelerate translational application and reduce development costs. Further studies using human-derived tissue models and clinical validation are warranted to confirm its clinical utility.

## Figures and Tables

**Figure 1 ijms-26-08868-f001:**
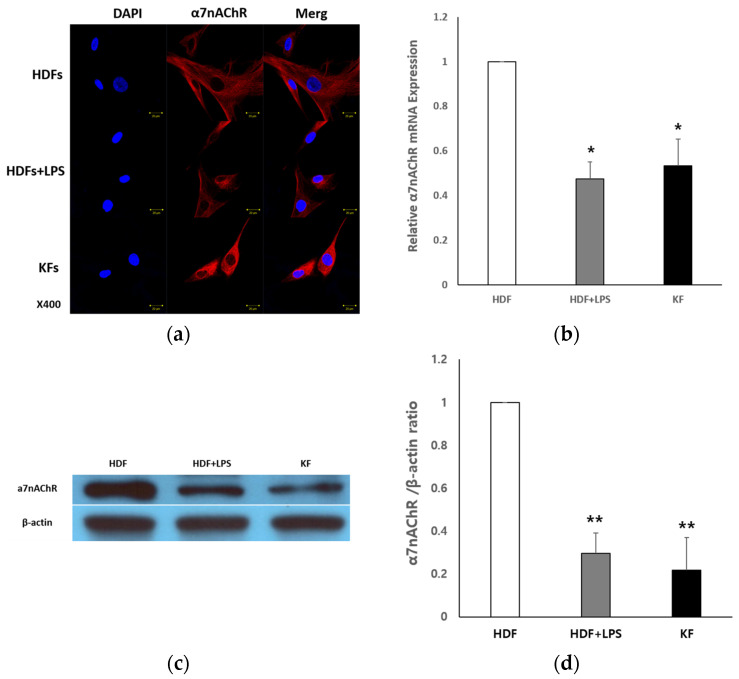
**α7nAChR expression in human dermal fibroblasts (HDFs) and keloid fibroblasts (KFs).** (**a**) Immunocytochemical staining for α7nAChR (red) and nuclei (blue, DAPI) in HDFs, lipopolysaccharide (LPS)-treated HDFs (100 ng/mL), and KFs. α7nAChR expression was markedly reduced in both LPS-treated HDFs and KFs. Scale bar = 20 μm. (**b**) Quantitative real-time PCR (qRT-PCR) analysis of α7nAChR mRNA expression. (**c**) Western blot analysis of α7nAChR protein in HDFs, HDF + LPS, and KFs. (**d**) Quantification of α7nAChR protein levels normalized to β-actin. Band intensities were quantified and compared among groups. α7nAChR mRNA and protein expression are significantly suppressed under inflammatory conditions and in KFs. Data are shown as mean ± SEM (n = 5). Statistical analysis was performed using repeated one-way ANOVA and paired t-test. * *p* < 0.05 for HDF vs. HDF + LPS and HDF vs. KF (qRT-PCR); ** *p* < 0.01 for HDF vs. HDF + LPS and HDF vs. KF (Western blot).

**Figure 2 ijms-26-08868-f002:**
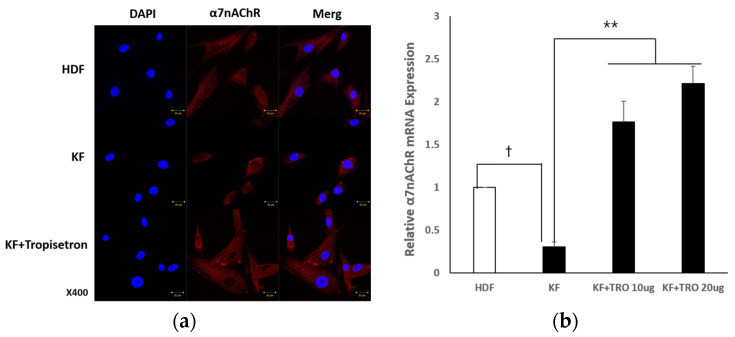
**Tropisetron enhances α7nAChR expression in keloid fibroblasts.** (**a**) Immunocytochemistry images showing α7nAChR (red) and DAPI-stained nuclei (blue) in HDFs, KFs, and KFs treated with 20 μg/mL tropisetron. For each condition, three separate panels are presented: DAPI (left), α7nAChR (middle), and merged image (right). Images were captured at 400× magnification. Red signals inside the cells may result from permeabilization, which allows antibody staining of both membrane and intracellular α7nAChR, including that present in the ER or Golgi. Scale bar = 20 μm. (**b**) qRT-PCR analysis of α7nAChR mRNA in HDFs, KFs, and KFs treated with tropisetron (20 or 40 μg/mL) for 48 h. Data are presented as mean ± SEM (*n* = 5). Statistical analysis was performed using repeated one-way ANOVA and paired t-test. † *p* < 0.05 (KF vs. HDF); ** *p* < 0.0001 (TRO-treated KF vs. untreated KF).

**Figure 3 ijms-26-08868-f003:**
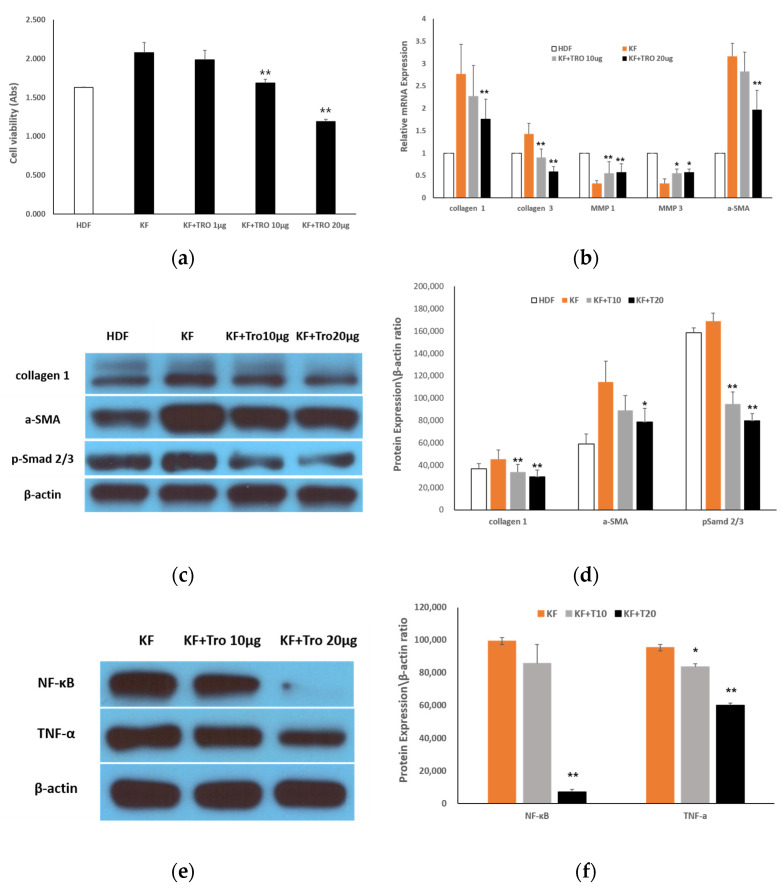
**Effects of tropisetron on keloid fibroblast viability and fibrotic/inflammatory markers.** (**a**) Cell viability assessed by MTT assay in KFs treated with various concentrations of tropisetron (1, 10, and 20 μg/mL) for 48 h. Cell viability decreased in a dose-dependent manner. (**b**) qRT-PCR of fibrosis-related genes (COL1A1, COL3A1, α-SMA) and matrix metalloproteinases (MMP1, MMP3) in HDFs, KFs, and tropisetron-treated KFs. Tropisetron reduced fibrotic gene expression and upregulated MMPs in a dose-dependent manner. (**c**) Western blot analysis of collagen I, α-SMA, p-Smad2/3 in HDFs, KFs, and tropisetron-treated KFs (10 and 20 μg/mL). (**d**) Densitometric quantification of Western blot bands in (**c**), normalized to β-actin. (**e**) Western blot analysis of NF-κB and TNF-α expression in KFs and tropisetron-treated KFs (10 and 20 μg/mL). (**f**) Densitometric quantification of bands in (**e**), normalized to β-actin. Tropisetron suppressed inflammatory markers in a dose-dependent manner. Data are presented as mean ± SD (*n* = 5). Statistical analysis was performed using repeated one-way ANOVA and paired t-test. (* *p* < 0.05, ** *p* < 0.01).

**Figure 4 ijms-26-08868-f004:**
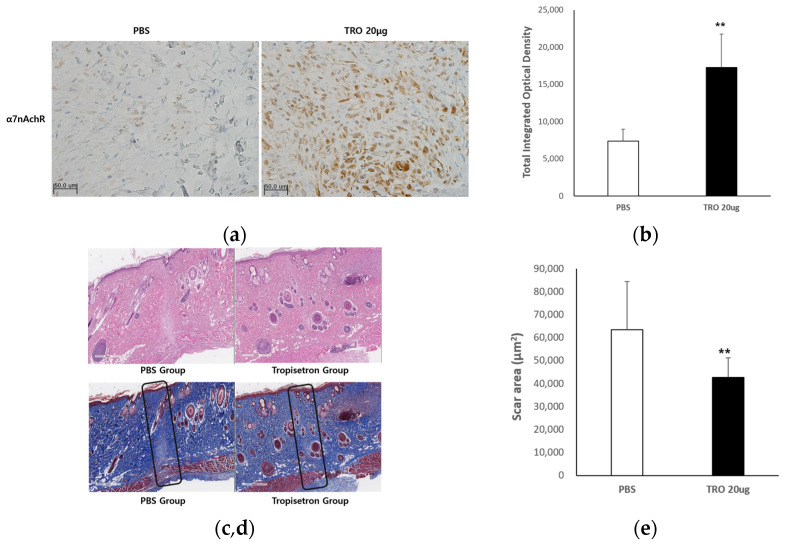
**Histological analysis of tropisetron-treated scars in vivo.** (**a**) Representative immunohistochemical staining images of α7nAChR in scar tissues from PBS-treated (control) and tropisetron-treated (20 μg) rats on day 14 post-incision. The tropisetron-treated group exhibited enhanced α7nAChR expression in the dermal layer compared to the PBS-treated control. Scale bar = 50 μm. (**b**) Quantification of α7nAChR-positive staining using MetaMorph® image analysis software (Universal Image Corp., Buckinghamshire, UK) (**c**) On H&E-stained images, the control group exhibited pronounced inflammatory infiltration and disorganized tissue structure, whereas the tropisetron-treated group showed reduced inflammation and more organized dermal architecture. Scale bar = 500 μm. (**d**) Masson’s trichrome (MT)-stained sections demonstrating dense collagen deposition in the control group and reduced, loosely arranged collagen fibers in the tropisetron-treated group. Scale bar = 500 μm. (**e**) Quantification of scar area measured from MT-stained sections. The scar area was significantly smaller in the tropisetron-treated group. Data are presented as mean ± SEM (n = 5). Statistical analysis was performed using repeated one-way ANOVA and paired t-test (** *p* < 0.01).

**Figure 5 ijms-26-08868-f005:**
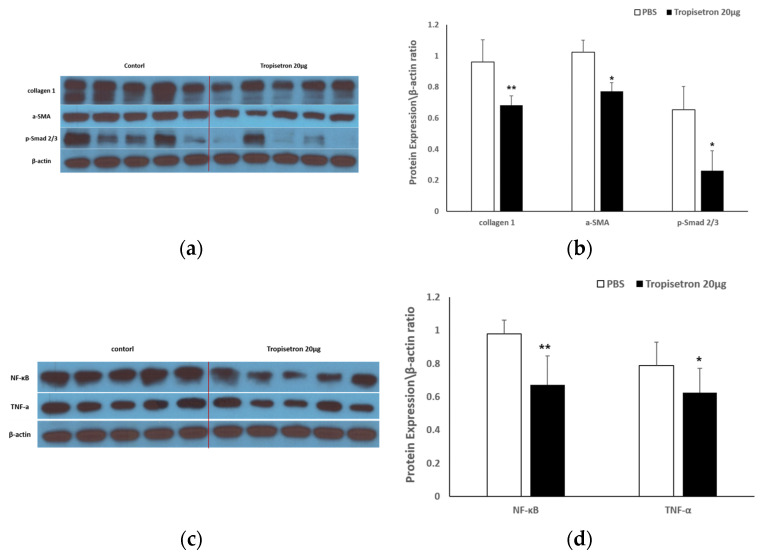
**Tropisetron reduces fibrosis and inflammation in scar tissue in vivo.** (**a**) Western blot analysis of collagen I, α-SMA, and p-Smad2/3 in scar tissues from PBS-treated (control) and tropisetron-treated (20 μg) rats at postoperative day 14. (**b**) Western blot analysis of inflammatory NF-κB and TNF-α in the same tissue samples. (**c**) Densitometric quantification of collagen I, α-SMA, and p-Smad2/3 expression normalized to β-actin. (**d**) Densitometric quantification of NF-κB and TNF-α expression normalized to β-actin. Data are presented as mean ± SEM (*n* = 5 per group). Statistical analysis was performed using repeated one-way ANOVA and paired t-test. * *p* < 0.05, ** *p* < 0.01, vs. PBS-treated group.

**Figure 6 ijms-26-08868-f006:**
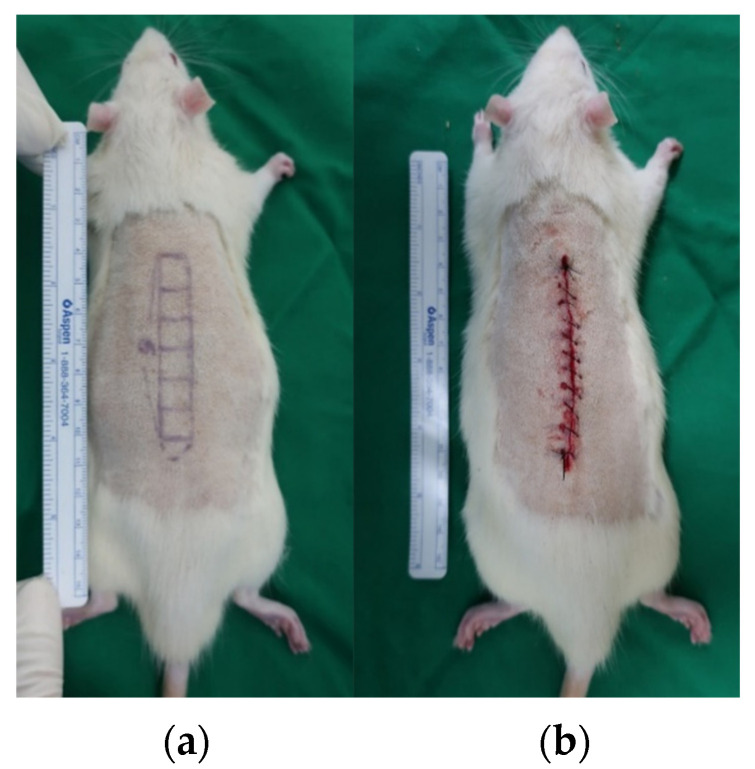
**Rat incisional scar model.** (**a**) Full-thickness rectangular incision (6 cm × 1 cm) on the dorsal skin. (**b**) Only the skin was sutured, leaving the muscle layer open to maximize tensile stress at the wound site.

## Data Availability

The data that support the findings of this study are available on request from the corresponding author. The data are not publicly available due to privacy or ethical restrictions.

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
