# Peer review of "Antifibrotic Effects of an α7 Nicotinic Acetylcholine Receptor Agonist in Keloid Fibroblasts and a Rat Scar Model"

_ijms, 2025, doi:10.3390/ijms26188868_

Round 1
Reviewer 1 Report
Comments and Suggestions for Authors
Dear authors,
Keloid research and the search for clearer targets and treatment options is very much engaging currently. Thanks for providing yet another option to plunder on to advance keloid disease research.
You have presented very useful options in nicotinic acetylcholine receptor (nAChR), particularly the α7 subtype (α7nAChR) and its agonist tropisetron, potential regulator of inflammation and fibrosis.
The manuscript has been well written, however raises a few questions that enhance the readability and comprehension of the paper if addresses.
Introduction
Page 2 of 15, Lines 48-50: I do understand the authors have demonstrated with immunocytochemistry & Western blot analysis, but apart from these literature cited which is not based on any discovery techniques such as NGS-Proteomics, transcriptomics etc., still highlighting effects of a7nAChR receptor on various fibrotic & inflammatory signaling pathways, are there any discovery data datasets, showing low expression of a7nAChR expression in KF...?
Page 4 of 15, Lines 110-111 (Figure 2a): Incomplete figure image...!; I was expecting to see three IF images, but can only see one, the cytoplasm & nuclei merged...?
Thank you.
Author Response
Comment 1:
Page 2 of 15, Lines 48-50: I do understand the authors have demonstrated with immunocytochemistry & Western blot analysis, but apart from these literature cited which is not based on any discovery techniques such as NGS-Proteomics, transcriptomics etc., still highlighting effects of a7nAChR receptor on various fibrotic & inflammatory signaling pathways, are there any discovery data datasets, showing low expression of a7nAChR expression in KF...?
Response:
Thank you for this thoughtful and insightful comment. We fully agree that transcriptomic or proteomic datasets can serve as valuable external evidence to support our findings. However, to the best of our knowledge, no publicly available transcriptomic or proteomic datasets (such as GEO, GTEx, or HPA) have reported α7nAChR (CHRNA7) expression levels in keloid fibroblasts. Therefore, our study provides the first experimental evidence of α7nAChR downregulation in KFs, as demonstrated by immunocytochemistry, qRT-PCR, and Western blotting (Figure 1). We have clarified this point in the Introduction with the following statement:
“To date, no transcriptomic or proteomic datasets have reported α7nAChR (CHRNA7) expression levels in keloid fibroblasts. In this study, we provide the first experimental evidence of α7nAChR downregulation in KFs using immunocytochemistry, qRT-PCR, and Western blotting. This novel finding has also been filed as a national patent, supporting its originality and potential therapeutic relevance.”
Comment 2:
Page 4 of 15, Lines 110-111 (Figure 2a): Incomplete figure image...!; I was expecting to see three IF images, but can only see one, the cytoplasm & nuclei merged...?
Response:
Thank you for pointing this out. Figure 2a includes three separate panels for each condition: DAPI (left), α7nAChR (middle), and merged image (right). We have revised the figure legend to clearly describe this structure.
“For each condition, three separate panels are presented: DAPI (left), α7nAChR (middle), and merged image (right).”

Reviewer 2 Report
Comments and Suggestions for Authors
The manuscript “Antifibrotic Effects of an α7 Nicotinic Acetylcholine Receptor 2 Agonist in Keloid Fibroblasts and a Rat Scar Model” of Hyun Roh et al is devoted to search of new approaches for treatment of wound healing and particularly to fight with keloids and scars, which often are supplemented with inflammation. As a new target for wound healing with antifibrosis effect the authors consider alpha7 nicotinic receptor.
The study is very interesting, valuable and is in demand. Nevertheless, some improvement is required an several points should be addressed:
- As the Methods section is placed at the end of the manuscript, please add short methodological description into the Results section in each new subpart to understand how and what experiments were performed.
- Localization of a7-nAChR expression in HDFs and KFs is not clear. It is important point, because tropisetron is a soluble drug (please add some short description of its properties and chemical formula into the manuscript). Thus, it can interact only with the receptor expressed on the cell surface. But from the images provided in the manuscript, it seems like the major part of the receptor is accumulated inside the cells. So, taking in mind that a7-nAChR is not a single target of tropisetron, it is strongly required to prove that tropisetron targets just a7-nAChR in HDFs and KFs. It can be knockdown experiment, or inhibition by alpha-bungarotoxin, or any other way. Analysis of a7-nAChR expression in the membrane fraction of the cells is also required.
- What is beta-action in each Y axis I the figures?
- Please, increase font of the legends in the figures
- Add description of statistical methods used in each figure legend.
- References in the first paragraph of the Discussion section are too old (> 20 years old, line 238). Please check the literature and add new references.
- Recently several works describing participation of nicotinic receptors and human endogenous regulatory protein SLURP-2 in wound healing were reported (Shlepova OV, et al. Peptide Mimicking Loop II of the Human Epithelial Protein SLURP-2 Enhances the Viability and Migration of Skin Keratinocytes. Acta Naturae. 2024, 16(4):86-94. doi: 10.32607/actanaturae.27494; L. Bychkov, et al, Human Epithelial Protein SLURP-2 as a Prototype of Drugs for Wound Healing. Rus. J. Bioorg. Chem., 2024, 50(3), 696–705. doi.org/10.1134/S1068162024030014). I suggest to include them into discussion and compare the signalling pathways involved in wound healing.
- Discussion section is full of logical and informative repeats. Please, rewrite this section by adding comparison with other works and approaches used for wound healing as well as by discussion of novelty and advantages of usage of a7-nAChR as a new target.
Author Response
Comment 1:
As the Methods section is placed at the end of the manuscript, please add short methodological description into the Results section in each new subpart to understand how and what experiments were performed.
Response:
Thank you for your thoughtful suggestion. In response, we have added a brief methodological description at the beginning of each Results subsection (Sections 2.1.1 to 2.2.2). These additions concisely summarize the experimental approach, including cell or animal models, treatments, and analytical techniques used. Full methodological details remain in the Methods section as per journal structure.
Comment 2:
Localization of a7-nAChR expression in HDFs and KFs is not clear. It is important point, because tropisetron is a soluble drug (please add some short description of its properties and chemical formula into the manuscript). Thus, it can interact only with the receptor expressed on the cell surface. But from the images provided in the manuscript, it seems like the major part of the receptor is accumulated inside the cells. So, taking in mind that a7-nAChR is not a single target of tropisetron, it is strongly required to prove that tropisetron targets just a7-nAChR in HDFs and KFs. It can be knockdown experiment, or inhibition by alpha-bungarotoxin, or any other way. Analysis of a7-nAChR expression in the membrane fraction of the cells is also required.
Response:
We thank the reviewer for this insightful comment regarding α7nAChR localization and the target specificity of tropisetron.
Regarding intracellular α7nAChR staining: We agree that the localization of α7nAChR is a key aspect. As suggested, we clarified this point in the Results section (2.1.2) and the Figure 2 legend by noting that the intracellular red signal may result from the permeabilization process used in our immunocytochemistry (ICC) protocol, which allows antibodies to access both surface and intracellular compartments. Additionally, α7nAChR has been reported to localize to intracellular organelles such as the endoplasmic reticulum or Golgi in non-neuronal cells [15], which may explain the cytoplasmic signals. Regarding the soluble nature and receptor selectivity of tropisetron: We have included a short description of the physicochemical properties of tropisetron in the Methods section (Section 4.1.2), noting that tropisetron is a water-soluble drug (C₁₈H₂₁N₃O₂) and functions as a selective partial agonist of α7nAChR in addition to being a 5-HT₃ receptor antagonist. We agree that its solubility supports interaction with surface receptors.
Regarding receptor specificity and lack of functional confirmation: We acknowledge the absence of knockdown or α-bungarotoxin inhibition experiments in our current study. To address this, we added a discussion in the manuscript’s final paragraph acknowledging this limitation and proposing that future studies include α7nAChR-specific inhibition (e.g., siRNA or α-bungarotoxin) and membrane fraction analyses to further validate receptor-specific effects of tropisetron.
We hope these revisions and clarifications adequately address the reviewer’s concerns.
Comment 3:
What is beta-action in each Y axis I the figures?
Response:
We assume the reviewer meant “β-actin” as this term appears in all relevant figure legends. In all figures, the Y-axis represents the relative protein expression levels normalized to β-actin, which was used as an internal control in the Western blot analyses. This normalization method was also described in the Methods section (Section 4.1.6).
Comment 4:
Please, increase font of the legends in the figures
Response:
Thank you for your suggestion. We have increased the font size of the figure legends in all figures to improve readability as requested.
Comment 5:
Add description of statistical methods used in each figure legend.
Response:
We thank the reviewer for this helpful suggestion. In response, we have revised all figure legends to include a brief description of the statistical methods used for data analysis. Specifically, we have indicated that data are presented as mean ± SEM and that statistical comparisons were performed using one-way ANOVA followed by Tukey’s post hoc test, or Student’s t-test, as appropriate. These details have been added at the end of each figure legend to clarify the statistical approach used for each experiment.
Comment 6:
References in the first paragraph of the Discussion section are too old (> 20 years old, line 238). Please check the literature and add new references.
Response:
Thank you for your valuable suggestion. We agree that the reference previously cited as [3] (Niessen et al., 1999) is outdated. Accordingly, we have replaced it with a more recent and comprehensive review article:
Ogawa, R. The Most Current Algorithms for the Treatment and Prevention of Hypertrophic Scars and Keloids: A 2020 Update of the Algorithms Published 10 Years Ago. Plast. Reconstr. Surg. 2022, 149, 79e–94e. https://doi.org/10.1097/PRS.0000000000008667.
This updated citation supports the same statement and reflects the most current understanding of keloid pathology and management.
Comment 7:
Recently several works describing participation of nicotinic receptors and human endogenous regulatory protein SLURP-2 in wound healing were reported (Shlepova OV, et al. Peptide Mimicking Loop II of the Human Epithelial Protein SLURP-2 Enhances the Viability and Migration of Skin Keratinocytes. Acta Naturae. 2024, 16(4):86-94. doi: 10.32607/actanaturae.27494; L. Bychkov, et al, Human Epithelial Protein SLURP-2 as a Prototype of Drugs for Wound Healing. Rus. J. Bioorg. Chem., 2024, 50(3), 696–705. doi.org/10.1134/S1068162024030014). I suggest to include them into discussion and compare the signalling pathways involved in wound healing.
Response:
We appreciate the reviewer’s valuable suggestion. As advised, we have incorporated recent findings regarding SLURP-2 and its role in wound healing into the Discussion section. These studies highlight the role of endogenous nicotinic receptor ligands such as SLURP-2 in enhancing keratinocyte function and re-epithelialization. We have compared these mechanisms with the α7nAChR-mediated actions of tropisetron, and added a comparative perspective on their convergent roles in wound healing and dermal remodeling. The corresponding references have also been added to the reference list.
(Discussion, lines 295-303)
“In addition to α7nAChR, other endogenous cholinergic regulators such as SLURP-2 have been implicated in cutaneous wound healing. Recent studies reported that SLURP-2 mimetic peptides enhance keratinocyte viability and migration, contributing to re-epithelialization and tissue regeneration through nicotinic receptor-mediated signaling [34,35]. Although the cellular targets of SLURP-2 may differ from tropisetron, both agents appear to share convergent pathways involving nicotinic receptor activation and wound repair facilitation. Further comparative studies are warranted to delineate distinct versus overlapping roles of SLURP-2 and α7nAChR-targeting agents in dermal remodeling.”
[34] Shlepova, O.V.; Parfenova, A.A.; Andreeva, N.V.; Shulepko, M.A.; Dolgikh, D.A.; Tsetlin, V.I.; Kirpichnikov, M.P. Peptide Mimicking Loop II of the Human Epithelial Protein SLURP-2 Enhances the Viability and Migration of Skin Keratinocytes. Acta Naturae 2024, 16, 86–94. https://doi.org/10.32607/actanaturae.27494.
[35] Bychkov, L.; Shlepova, O.V.; Dolgikh, D.A.; Kirpichnikov, M.P.; Tsetlin, V.I. Human Epithelial Protein SLURP-2 as a Prototype of Drugs for Wound Healing. Russ. J. Bioorg. Chem. 2024, 50, 696–705. https://doi.org/10.1134/S1068162024030014.
Comment 8:
Discussion section is full of logical and informative repeats. Please, rewrite this section by adding comparison with other works and approaches used for wound healing as well as by discussion of novelty and advantages of usage of a7-nAChR as a new target.
Response:
We thank the reviewer for this insightful comment. In response, we have revised the Discussion section to clarify the distinct roles of α7nAChR in comparison to other wound healing mediators, including recent studies on SLURP-2-mediated mechanisms (Refs. [34,35]). We also highlighted the novelty of α7nAChR as a therapeutic target in pathological scarring and the clinical repurposing potential of tropisetron. While some reiteration was retained for summarizing in vitro and in vivo findings, we have minimized redundancy and ensured each paragraph delivers distinct insights.

Round 2
Reviewer 1 Report
Comments and Suggestions for Authors
Dear Authors,
Thanks for the responses to issues raised, I still do think some NGS data would have strengthen your claim which clearly we all allude to the fact that this data is not available in the already published literature.
Best wishes
Author Response
Comment :
Dear Authors,
Thanks for the responses to issues raised, I still do think some NGS data would have strengthen your claim which clearly we all allude to the fact that this data is not available in the already published literature.
Best wishes
Response:
Thank you very much for your thoughtful follow-up comment.
We fully agree that the inclusion of transcriptomic or NGS-based data would further strengthen the mechanistic insights of our study. However, as you rightly noted, such discovery-level datasets specifically addressing α7nAChR expression in keloid fibrosis remain unavailable in the existing literature.
We hope that our findings will serve as a valuable foundation for future high-throughput investigations exploring this novel therapeutic target.
Once again, we appreciate your constructive feedback and the opportunity to improve our work.
Reviewer 2 Report
Comments and Suggestions for Authors
There are typos in Y axes of majority of figures: besides b-action should be b-actin!
Author Response
Comment :
There are typos in Y axes of majority of figures: besides b-action should be b-actin!
Response:
Thank you for bringing this to our attention.
We apologize for the typographical error in the Y-axis labels of several figures. The term “b-action” was inadvertently written instead of the correct “β-actin.” We have carefully reviewed and corrected this labeling error in all relevant figures.
We greatly appreciate your careful review and attention to detail.
Round 3
Reviewer 1 Report
Comments and Suggestions for Authors
Dear Authors,
Thanks for your enthusiasm on this, we all looking forward to a more robust validation, however, as rightly highlighted, this manuscript would definitely set a base for future confirmations.
Best wishes!
Author Response
Comments 1 :
Dear Authors,
Thanks for your enthusiasm on this, we all looking forward to a more robust validation, however, as rightly highlighted, this manuscript would definitely set a base for future confirmations.
Best wishes!
Response 1 :
We are sincerely grateful to the reviewer for the kind and encouraging comments.
We fully agree with the reviewer’s perspective that the present study represents an initial step and provides a valuable foundation for future confirmations.
We greatly appreciate the reviewer’s recognition of the potential significance of our work, and we are committed to further validating and expanding upon these findings in subsequent studies.
Sincerely,
Won Jai Lee, M.D., Ph.D. (Corresponding Author)
on behalf of all authors